# Examination of the Productivity and Physiological Responses of Maize (*Zea mays* L.) to Nitrapyrin and Foliar Fertilizer Treatments

**DOI:** 10.3390/plants10112426

**Published:** 2021-11-10

**Authors:** Dalma Rácz, Lóránt Szőke, Brigitta Tóth, Béla Kovács, Éva Horváth, Péter Zagyi, László Duzs, Adrienn Széles

**Affiliations:** 1Faculty of Agricultural and Food Sciences and Environmental Management, Institute of Land Use, Engineering and Precision Farming Technology, University of Debrecen, 138 Böszörményi St., 4032 Debrecen, Hungary; racz.dalma@agr.unideb.hu (D.R.); horvath.eva@agr.unideb.hu (É.H.); zagyi.peter@agr.unideb.hu (P.Z.); duzs.laszlo@agr.unideb.hu (L.D.); szelesa@agr.unideb.hu (A.S.); 2Faculty of Agricultural and Food Sciences and Environmental Management, Institute of Food Science, University of Debrecen, 138 Böszörményi St., 4032 Debrecen, Hungary; btoth@agr.unideb.hu (B.T.); kovacsb@agr.unideb.hu (B.K.)

**Keywords:** abiotic stress, foliar fertilizer, maize, nitrapyrin, nutrient supply, stress response

## Abstract

Nutrient stress has been known as the main limiting factor for maize growth and yield. Nitrapyrin, as a nitrification inhibitor—which reduces nitrogen loss—and foliar fertilizer treatments have been successfully used to enhance the efficiency of nutrient utilization, however, the impacts of these two technologies on physiological development, enzymatic responses, and productivity of maize are poorly studied. In this paper, the concentration of each stress indicator, such as contents of proline, malondialdehyde (MDA), relative chlorophyll, photosynthetic pigments, and the activity of superoxide dismutase (SOD) were measured in maize leaf tissues. In addition, biomass growth, as well as quantitative and qualitative parameters of yield production were examined. Results confirm the enhancing impact of nitrapyrin on the nitrogen use of maize. Furthermore, lower activity of proline, MDA, SOD, as well as higher photosynthetic activity were shown in maize with a more favorable nutrient supply due to nitrapyrin and foliar fertilizer treatments. The obtained findings draw attention to the future practical relevance of these technologies that can be implemented to enhance the physiological development and productivity of maize. However, this paper also highlights the importance of irrigation, as nutrient uptake from soil by the crops decreases during periods of drought.

## 1. Introduction

Climate change and its adverse effects on maize production have been widely studied [1,2,3,4,5]. Global warming contributes to more frequent drought and heat stress leading to a significant reduction in yield quality and quantity [6]. However, an increasing number of researchers examining the impacts of abiotic stress on crop health published that nutrient stress has the greatest impact on growth [7]. Increased emphasis must be put on studies addressing the effects of nutrient deficiencies since appropriate and balanced nutrition supports maize health and its tolerance to abiotic stress [8]. Thus, in recent years, attention has increased to research focusing on nutrient-enhancing technologies.

Maize development, biomass production, and yield are largely determined by the amount of available nitrogen (N) in soils [9] since it is the most limiting nutrient for growth [10]. Considering N losses, N fertilizers are ideal to be applied at later stages of maize development, when the efficiency of the absorption and the use of N by the plant increases [11]. Although maize requires substantial N input, it has been shown that excessive N fertilization does not increase yield, but it contributes to increased N loss [12,13]. As prior studies have already revealed, substantial N loss results mainly from nitrate (NO_3_^−^) leaching from the soil surface [14,15] which contaminates the quality of groundwater [16,17,18]. Thus, it is struggling to ensure optimal N fertilizer for crops that must be under strict environmental regulations. Additionally, the utilization of N is affected by the weather. Heavy rainfalls or higher temperatures heighten the risk of losing N from leaching or volatilization, posing a challenge to growers deciding on the amount of N to be applied [19].

Nitrapyrin [2-chloro-6-(trichloromethyl)-pyridine] as an ingredient of nitrification inhibitor is a promising strategy to enhance the efficiency of N utilization without excess N fertilization since it reduces N loss and contributes to providing more available N in the soil [20,21]. Nitrapyrin inhibits a specific population of nitrification bacteria (*Nitrosomonas* sp.) which convert ammonium ions to nitrite ions through the inhibition of the ammonia monooxygenase (AMO). Hence, the stable and less mobile ammonium (NH_4_^+^) forms will be available for a longer period in the soil, and nitrate leaching decreases, resulting in enhanced N utilization and improved yield production of the crops [21].

Recently, this N stabilizer has gained more attention due to its environmental benefits ranging from decreased nitrate leaching and nitrous oxide emission to increased yield [22,23,24].

Another strategy to support maize health conditions is a complementary foliar fertilizer treatment. Foliar fertilization of arable crops has not been a part of everyday practice [25], and it has been used traditionally to make up for nutrient deficiencies [26]. However, weather extremities—such as heat stress and drought—and the limited availability of micronutrients in the soils prompted farmers to provide extra support to crops by supplementing soil fertilization [27]. Particularly strong need for foliar feeding emerges during soil water stress when nutrient uptake becomes energetically unfavorable for each plant [26], as well as during periods of rapid growth [28]. However, it must be noted that foliar fertilization can be used merely as a supplemental method and does not replace fertilization of the soil with the major nutrients [29]. As many prior studies have confirmed, foliar fertilizer treatments can improve nutrient use efficiency, heal nutrient deficiency, mitigate the harmful effects of stress on crops [30,31], and enhance photosynthetic activity. Additionally, it contributes to achieving higher yield quality and quantity [32].

Abiotic stressors cause oxidative stress in crops which damages vital macromolecules such as lipids, proteins, and nucleic acids [33]. Considering plant physiology, one of the greatest damages is caused by cell damage due to the peroxidation of lipids on cell membranes [34]. To mitigate such adverse conditions, crops have evolved an antioxidant defense system which also allows the analytical study of the effects of environmental factors through the activity of the enzymatic elements of this system [35]. Insufficient N supply (both excess and low rate) as an abiotic stressor triggers the accumulation of reactive oxygen species (ROS), resulting in oxidative stress in crops. In addition, excessive ROS can react with key photosynthetic substances processing chlorophyll degradation [36]. The first-line defense against ROS accumulated in plant cells due to damaging oxidative stress effects is provided by, inter alia, metal-containing superoxide dismutase (SOD). Its increased level due to stress helps to neutralize reactive oxygen species [37], thus reducing adverse effects in crops [38,39]. It has been revealed that the activities and expression of antioxidant enzymes such as SOD are strongly linked to N supply [40] as it is significantly increased with N fertilization up to a certain range [41]. This phenomenon is consistent with a study on nitrapyrin—which aims to optimize N utilization—that increases in the photosynthetic activity by providing more N and enhanced activity of superoxide dismutase (SOD) were observed in waterlogged maize [42]. In addition, the accumulation of proline as a “stress amino acid” in plant cells is a widespread response to adverse (especially water deficiency and excessive salt concentration) environmental factors [43,44]. Although the increased N supply of maize by adding nitrapyrin to N fertilizer does not directly influence the content of proline in leaf tissues [45], maize with an improved condition resulting from enhanced N and other nutrients supply is likely to be more tolerant to abiotic stress.

This study aims to obtain deeper knowledge about the correlation between various nutrition forms of maize resulting from the N stabilizer (nitrapyrin) and foliar fertilizer treatments in the field, as well as the tissue concentration of each stress indicator. The authors believe that all plants that have not been treated with a combination of the two technologies will develop nutrient stress due to their poorer nutrient supply. To monitor the efficiency of nitrapyrin, nitrate content was measured regularly in the soil. In this present research, the stress response to different nutrition of maize was studied by measuring the contents of proline, the rate of lipid peroxidation, relative chlorophyll, and photosynthetic pigments (chlorophyll *a*, chlorophyll *b*, and carotenoids), as well as the activity of superoxide dismutase (SOD). Since foliar feeding is attracting renewed attention due to global warming, and nitrapyrin has gained considerable interest in the practical use, we believe this study advances in the knowledge of the effect of these two technologies on stress adaptation of maize studied enzymatically, which was unprecedented before, thereby obtaining encouraging results.

## 2. Results

### 2.1. The Efficiency of Nitrapyrin–Nitrate Content in the Soil

The efficiency of soil-applied nitrapyrin was monitored by a regular measurement of nitrate content changes in nitrapyrin-treated (NP) and untreated (CT) soil which was paired with soil temperature data. Results suggest that the inhibitory effect of nitrapyrin on nitrification was significant until the seventh week after the nitrapyrin treatment, as the nitrate content in the untreated (CT) soil was much lower than in the nitrapyrin-treated plots (NP). In addition, the obtained results indicate that the inhibitory effect of nitrapyrin decreased with a considerable rise in soil temperature, which began on the seventh week after the application. No difference was observed between the results of the measurements on the ninth, eleventh, and fourteenth weeks after application (Figure 1).

### 2.2. Photosynthetic Parameters

Determination of the photosynthetic parameters was based on the measurements of the relative chlorophyll value, as well as the quantity of the chlorophyll a, chlorophyll b and carotenoid pigments individually. As for the relative chlorophyll content measurements, results obtained at the V12 stage showed that there was a significant difference between the control (CT) and various other treatments (NP, NP + FF). However, no difference was observed between the nitrapyrin-treated maize (NP) and the joint treatment of nitrapyrin and foliar fertilizer (NP + FF). Similar results were obtained in the second stage measured at stage R1, however, the joint treatment (NP + FF) resulted in the highest SPAD readings in the leaves. In summary, the increase in relative chlorophyll content was mainly influenced by nitrapyrin (NP) (Figure 2).

By examining the concentration of the individual photosynthetic pigments (chlorophyll *a*, chlorophyll *b*, carotenoid) in leaf tissues, it was found that, at stage V12, only the nitrapyrin treatment (NP) contributed to a significant increase of chlorophyll *a* (Chl *a*), while chlorophyll *b* (Chl *b*) was not influenced by any of the treatments. In the case of the ratio of Chl *a* to Chl *b*, there was a minor, but not a major difference between the treatments. At the V12 stage, carotenoid (Car) contents were not affected by any treatment. The total chlorophyll/carotenoid ratio increased mostly due to nitrapyrin (NP).

Results obtained at the R1 stage suggest that Chl *a* concentration increased due to all the three treatments (NP, FF, NP + FF), particularly the combined treatment (NP + FF). In the case of Chl *b*, only the combined treatment (NP + FF) resulted in higher concentrations. The ratio of Chl *a* to Chl *b* was still not significantly affected by any treatment. In terms of carotenoid content, a slight increase was observed due to foliar fertilizer treatment (FF), and an increase was shown due to the joint treatment (NP + FF). A similar tendency was observed in terms of the total chlorophyll/carotenoid ratio as it increased chiefly in nitrapyrin-treated (NP) maize (Table 1).

### 2.3. Stress Parameters

Here we present results of the concentration of malondialdehyde (MDA), the activity of superoxide dismutase (SOD), and the content of proline as stress indicators in maize leaf tissues. For the MDA concentration, the differences between treatments had been already shown at stage V12, as the highest MDA level values were gained in the control treatment (CT) and a decrease was observed in the joint treatment (NP + FF). Although the results obtained in stage R1 suggest that the difference between the three treatments (NP, FF, NP + FF) is moderated, the MDA concentration in the control (CT) maize remained significantly higher (Figure 3).

As regards the SOD activity, results obtained at the V12 stage already revealed a highly significant increase in the SOD activity level in untreated (CT) cultivars, which was reduced in all other treatments (NP, FF, NP + FF). At the R1 stage, all treatments (NP, FF, NP + FF) resulted in a decrease in SOD activity. However, the best results were obtained by the joint treatment of nitrapyrin and foliar fertilizer (NP + FF) (Figure 4).

Regarding the proline content, the results obtained at the V12 stage revealed that the highest proline content was achieved in the untreated (CT) maize which decreased in all other treatments (NP, FF, NP + FF). The proline content was the lowest due to foliar fertilization (FF) treatment and the joint application of nitrapyrin and foliar fertilizer (NP + FF), however, no significant difference was observed between the two. Results obtained at the R1 stage show a similar tendency as the three treatments (NP, FF, NP + FF) contributed to the substantial decrease of proline content. The lowest proline contents were observed in the nitrapyrin treatment (NP) and the combination treatment of nitrapyrin and foliar fertilizer (NP + FF) (Figure 5).

### 2.4. Nutritional Leaf Analysis

The actual nutrient content of maize leaves depending on the performed treatments (CT, NP, FF, NP + FF) is presented in Table 2. The obtained results suggest that the N content of the maize crops regardless of the applied treatments was below the critical level (28,000 mg kg^−1^ [46]), i.e., maize crops suffered from N deficiency. However, N content slightly increased in maize treated with both nitrapyrin and foliar fertilizer (NP + FF), which can be explained by the extra N content that was applied in the form of foliar fertilization (80 mg L^−1^). Although many of the nutrients were below the critical level (N, P, K, S), a slight increase was observed in maize treated with NP + FF. At the same time, the authors are convinced that, due to lack of precipitation resulting from the adverse weather conditions, these nutrients could not be adequately utilized by the crops (only 3.4 mm precipitation fell on the ninth week after maize sowing when leaf samples were collected; Figure 6). For this reason, attention is also drawn to the adverse effects of climate change, as the nutrient use efficiency of maize decreases under non-irrigated conditions or during periods of drought [47].

### 2.5. Biomass Growth

Due to nitrapyrin-containing treatments (NP, NP + FF), both stem diameter and root mass of maize were increased considerably compared to untreated maize (CT), which suggests that nitrapyrin contributed to retaining more available N-forms in the soil, resulting in maize crops increasing their biomass production. The obtained results are presented in Figure 7.

### 2.6. Yield Parameters

The results of maize ear parameters are presented in Table 3. The obtained data suggest that beneficial changes were observed due to NP, FF, and NP + FF treatments in parameters including 1000 kernel weight, the length and diameter of the ear, the number of rows per ear, and the number of grains per row. However, results were significantly improved in NP- and NP + FF-treated maize crops, which implies that these agronomic yield parameters were mostly positively affected by the nitrapyrin treatment. Regarding yield quality parameters, starch content slightly increased mostly due to FF treatment, and protein content increased partially due to NP and NP + FF treatment, while the oil content did not increase with either treatment.

## 3. Discussion

In light of the outcomes, the efficiency of nitrapyrin is strongly influenced by soil temperature, as the difference between the nitrate content of nitrapyrin-treated (NP) and untreated (CT) soils has decreased with raising soil temperature, suggesting a reduction in the persistence of nitrapyrin. This result is consistent with previous studies which outlined that nitrapyrin started to become less effective when soil temperature elevated, as a result of decomposition leading to a reduction in the inhibition on nitrification [48,49,50]. A study by Touchton et al. published in 1979 [51]—not long after its first report of nitrapyrin in 1962 by Goring [52]—already highlighted that the application of nitrapyrin is not recommended when the soil temperature is above 13 °C, otherwise, it will result in unsatisfactory nitrification control. A similar result was published in a recent study by Byrne et al. (2020) [53] which stated that nitrapyrin is stable and persistent only in cool soils, but it goes under chemical hydrolysis in warmer soils within 30 days. The obtained findings are of direct practical relevance, i.e., nitrapyrin is advisable to be applied in the autumn season on colder soils to mitigate N loss, as the soil temperature—as an interacting factor—largely determines nitrapyrin’s mode of action [54]. In contrast, some previous studies found that spring-applied nitrapyrin effectively reduces N loss during wet vegetation as well [55]. Furthermore, Pittelkow et al. (2017) [56] reported the inefficiency of fall-applied nitrapyrin. At this point, the uneven precipitation distribution in this present experiment needs to be pointed out. The efficiency of nitrapyrin started to decline not only when the soil temperature rose, but also when sufficient precipitation was lacking, as well as in the time of drought. The authors’ observation is in correspondence with several studies which have shown that nitrapyrin sufficiently decreased N loss in the waterlogged field where nitrate leaching is a substantial source of N loss [42,55]. In our research, due to frequent precipitation, nitrate leaching has increased in the untreated soil causing lower nitrate content. Conversely, in the nitrapyrin-treated soil, despite the regular rainfall, nitrapyrin contributed to retaining more N. In other words, the amount of nitrate leached from the untreated soil by rainfalls could be utilized by the crops from the nitrapyrin-treated soil. Thus, based on the authors’ assumptions, the effect of nitrapyrin would have been prolonged if there had been abundant rainfall.

The higher relative chlorophyll content obtained in nitrapyrin-treated maize (NP) is explained by the fact that maize increases its yield due to the accessibility of different N forms in the soil provided by nitrapyrin [57]. Several prior studies have already supported the authors’ findings as nitrapyrin increases plant chlorophyll by providing nutrients and increasing photosynthesis capacity; therefore, it increases leaf SPAD values which indicate the N status of maize [58,59,60]. It is commonly known that the photosynthetic activity of maize is closely linked to its general health conditions [32,61]. For this reason, it was not a surprise that the highest SPAD values at the R1 stage were obtained in maize co-treated with nitrapyrin and foliar fertilizer (NP + FF). Lu et al. (2001) [62] also reported that insufficient N supply reduces photosynthesis. However, it is worth noting that data obtained in the present research show that cultivars treated with foliar fertilizer alone (FF) only slightly increased their SPAD values. A study by Ling et al. (2002) [29] can be an explanation for this finding, as their results suggest that foliar fertilization is only a supplemental method that may correct nutrient deficiencies but cannot replace soil-applied fertilizers of major nutrients. Thus, as chlorophyll molecules contain N [63], it can be concluded that the increase of the relative chlorophyll content was mainly due to more favorable N conditions in the soil provided by nitrapyrin (NP).

As reported by several prior studies, N deficiency always causes a decrease in leaf chlorophyll concentration, since the level of N supply has a positive correlation with the chlorophyll content of crops [64,65]. This finding is consistent with the present study, in which the Chl *a* content increased mostly in maize with improved N supply due to nitrapyrin. Although Peng et al. (2021) have recently reported that, in addition to Chl, Car contents also showed an upward trend with an increasing N application rate [66], the authors’ observation is that it cannot be concluded. However, at the R1 stage, the joint treatment of nitrapyrin and foliar fertilizer resulted in higher Car content, which suggests that the joint treatment of nitrapyrin and foliar fertilization (which also contained N) may have resulted in the best N supply conditions for maize, thereby increasing Car content Furthermore, in a study by Tóth et al. (2002) [67], the Car and Chl *a*/*b* ratio increased with the reduction in N supply. In contrast, Zhao et al. (2003) [61] reported a decrease in Car contents in N deficiency. Based on these controversial results, further study of the issue is still required.

The determination of MDA concentration as a product of peroxidation of the unsaturated fatty acids in phospholipids can indicate the free radical damage to cell membranes under stress conditions [68,69]. In the present research, nitrapyrin (NP) and foliar fertilizer (FF), and the combination of the two (NP + FF) prevented lipid peroxidation by reducing the MDA concentration in the leaves compared to the untreated cultivars (CT) at both sampling times. Nitrapyrin likely contributed to more favorable conditions of maize by providing more available N forms in the soil. As a consequence, the lack of N in untreated maize (CT) resulted in oxidative stress, increasing MDA concentration. The obtained results were confirmed in a recent study by Li et al. (2020) [70] who stated that N deficiency would lead to the breakdown of soluble protein and the increase of MDA concentration, which damages the protective enzyme system, thereby accelerating leaf aging. As reported by Jiang et al. (2005) [71], the increase of the applied N mitigates the rate of lipid peroxidation. Moreover, N fertilizer has a significant effect on the degree of lipid peroxidation [72]. A study on the efficiency of nitrapyrin also supports the obtained results, as it contributed to mitigating (waterlogging) damages on the antioxidant system by increasing the activities of protective system enzymes and decreasing the MDA concentration [42]. In this present experiment, foliar fertilization rich in N was also conductive to alleviating stress in maize, suggesting that fertilized plants were less affected by environmental stresses [71]. In addition, the obtained results suggest that foliar fertilization had a greater impact on MDA concentration compared to nitrapyrin, as the two lowest values were measured in co-treated (NP + FF) and foliar treated (FF) maize. Similar results were observed with foliar fertilization rich in N in barley [73] and rice [74] under water deficit stress.

In this study, SOD showed significantly lower activity in all treatments (NP, FF, NP + FF) compared to the control (CT) which suggest that more adequate nutrition of maize by providing more available N due to nitrapyrin treatment (NP) and other nutrients (FF) contributed to mitigate oxidative stress in crops. The obtained results showed that the better N supply reduced the SOD activity, which has been confirmed by prior studies on how the activity of SOD is affected under different N supply forms of maize. Yue et al. (2021) [19] found that excessive N fertilization (200 and 300 kg ha^−1^) increased substantially the SOD activity, while Yang et al. (2019) [75] reported that N deficiency also induced increased SOD activity which decreased after N resupply. Foliar feeding (FF) also improved the stress tolerance of maize, as SOD decreased compared to non-foliar fertilized cultivars (CT). This finding was likely because other nutrients such as iron (Fe), zinc (Zn), copper (Cu), and manganese (Mn) also play a key role as cofactors in the structure of many antioxidant enzymes [76,77]. These cofactors are required by SOD for effective ROS detoxification [78,79]. Thus, the lack of these elements contributes to the lower activity of the antioxidant enzymes, imposing increased sensitivity to environmental stresses [76]. As a consequence, it is clear that the lower SOD activity measured in foliar fertilized maize (FF) was a consequence of the alleviated stress conditions in crops.

According to the present knowledge about the function of proline, it is a multifunctional amino acid [80] that helps to stabilize sub-cellular structures (e.g., membranes and proteins), as well as to scavenge free radicals under stress [81]. Hence, proline is a widely used indicator that accumulates to high levels as a response to stresses [82], especially water deficit [83], salinity [84], and heavy metal accumulation [85]. Moreover, a study in 1979 by Göring et al. [86] already reported that proline also accumulates under nutrient deficiency such as N. Tarighaleslami et al. (2012) [87] also found that, during stress, proline as an N storage tank reduces the osmotic potential of the cytoplasm. This agrees with the results of this study concerning untreated maize (CT) and thus, N poor cultivars showed a higher level of proline in leaf tissues compared to the nitrapyrin-treated (NP) cultivars. Since studies on the effects of other microelements on proline accumulation are lacking presumably due to an irrelevant relationship between the two, it can be concluded that both nitrapyrin and nitrogenous foliar fertilizer improved the N supply of treated crops, creating more favorable conditions of stress control.

The efficacy of nitrapyrin treatment was also manifested in increased root mass and thicker stem diameter, which implies that nitrapyrin provided more available N forms in the soil, thereby, improving the N use efficiency of maize. [88]. For maize yield parameters, the best results were obtained in NP and NP + FF-treated maize crops, which suggests that mostly the nitrapyrin-containing treatments contributed to improving them. Ren et al. (2017) [55] also found that nitrapyrin significantly increased kernel number and 1000-kernel weight. In quality parameters, only slight changes were measured due to the different treatments. While a slight increase in protein content was due to nitrapyrin-containing treatments (NP, NP + FF), the starch content was mostly influenced by foliar fertilization (FF). A previous study on nitrapyrin by Singh and Nelson, 2019 [89] reported a similar result, in which nitrapyrin increased the protein content which could be explained by the higher grain N concentration since N is a pivotal component of protein. However, another study found a similar finding that foliar fertilization caused an increase in starch content. [90].

Although with different efficiency values for each applied treatment (NP, FF, NP + FF), the result of the nutritional laboratory leaf analysis was rather controversial as no significant improvement in nutrient content was observed with either treatment. The measured N, P, K, and S content was below the minimum critical value, and none of the performed treatments could make up for these hidden nutrient deficiencies. However, it must be pointed out that this result provides only the actual nutritional status of maize due to the single occasion of sampling. Furthermore, maize leaves were collected during the period of water stress, as a negligible amount of precipitation fell which is important to note because several studies have shown that, in plant tissues, drought potentially decreases nutrient uptake from the soil [47,91,92].

## 4. Conclusions

The expansion of maize production and long-term projections of climate change encourage researchers to study technologies that improve nutrient utilization, and, as a result, mitigate oxidative stress in maize caused by nutrient stress. In this paper, two yield-enhancing technologies, foliar fertilizer (containing N and other nutrients), and nitrapyrin contributed to providing more N to maize, which manifested in a significant decrease in the concentrations of stress indicators, such as SOD, MDA, proline, and in a significant increase in contents of relative chlorophyll and photosynthetic pigments. These findings draw attention to adequate N management since it has also a substantial impact on the biomass production of maize, and it improves the protein content of yield. Nitrapyrin is a promising yield-enhancing technology to improve the physiology and productivity of maize by enhancing N use. However, it must be noted that this paper presents a field study, in which adverse weather conditions, such as lack of precipitation certainly influenced the physiological development of maize. Irrigation should be indispensable for the enhanced nutrient utilization from the soil, as uneven rainfalls and prolonged droughts are to be expected in the future. For this reason, this issue needs further investigation.

## 5. Materials and Methods

### 5.1. Design of the Field Experiment, Treatments, and Sampling

The study site was located in the Demonstration Garden of the Institute of Land Use, Engineering and Precision Farming Technology (University of Debrecen, Debrecen, Hungary; 47°33′07.9″ N, 21°36′03.0″ E). All data were obtained in 2021. The field experiment represents four treatments (control: CT; nitrapyrin: NP; foliar fertilizer: FF; nitrapyrin + foliar fertilizer: NP + FF) which were repeated five times (randomized block design) in fodder maize (*Zea mays* L., FAO 490, Kite Zrt.). Nitrapyrin (N-Lock^TM^ Max, Corteva Agriscience, Wilmington, DE, USA) was sprayed (22 April; 1.7 L ha^−1^; in 300 L water ha^−1^) before maize sowing (27 April) and it was immediately incorporated into the soil (6–8 cm depth). Furthermore, its application was adapted to the incoming precipitation. Four mm rain fell within a few hours after spraying, and a further 12 mm precipitation within two weeks helped to activate the soil-applied nitrapyrin. Foliar fertilizer (10 L ha^−1^; in 300 L water ha^−1^) was applied on 23 June, at the stage of V9-V10 (BBCH 19), when maize had sufficient leaf area to absorb nutrients. The nutrient composition of foliar fertilizer is presented in Table 4.

Samples of the top fully expanded maize leaves were taken on two occasions (at stage V12, 11th week after sowing, 6 July; and maize silking (R1), 2 August 14th week after sowing) to examine the stress response of maize to different nutrient supplies resulting from nitrapyrin and foliar fertilizer treatments. These two stages were chosen because both periods are critical stages in maize development. At the V12 stage, the potential number of kernels on each ear and the ear size (length) were determined. Nutrients are critical during this period as maize is in a rapid growth phase. In addition, as a result of nutrient or water stress, the largest yield reduction occurred at the R1 stage (silking) [93], which is a part of the reproductive stages.

### 5.2. Soil Properties

Soil samples were collected from the experimental field on 9 March, before the growing season, and were analyzed by the accredited HL-LAB Environmental and Soil Testing Laboratory (Debrecen, Hungary). The leached chernozem-type soil profile has excellent properties. The plasticity index according to Arany (K_A_), which describes the soil texture, was 43 (clay loam) [94]. The average pH_KCL_ is 7.50 [95]. Humus content is 1.99%, carbonated lime content is 14.9%. The AL-soluble P_2_O_5_ content is 256 mg kg^−1^, the AL-soluble K_2_O is 162 mg kg^−1^ [96]. The KCl-soluble N-NO_3_ + NO_2_ (all nitrate + nitrite; [96]) content is 2.78 mg kg^−1^. After soil sampling, spring soil preparation with mineral fertilization was conducted with 90 kg ha^−1^ N, 23 kg ha^−1^ CaO, and 16 kg ha^−1^ MgO on 30 March.

### 5.3. Weather Conditions

In the study area, weekly precipitation and temperature data were collected from 27 April (maize sowing) to 2 August (last sampling date) by the weather station of the Demonstration Garden of the Institute of Land Use, Engineering, and Precision Farming Technology of the University of Debrecen. Measurements were taken using a WxPRO™ weather station (Campbell Scientific, Shepshed, UK). Data of soil temperature was collected with an SM150T soil moisture sensor (Delta-T Devices, Cambridge, UK). Although spring brought abundant rainfall which moistened the soils, even mid-April did not provide the proper conditions for timely maize sowing (27 April) due to the slow rise in temperature and frosts. Initial maize development was delayed and soils were wet for a prolonged period. There was enough precipitation to cover the water needs of maize (74 mm) until the fifth week after maize sowing. However, the temperature was still unusually low (average 14.7 °C). During the subsequent period, an extremely uneven distribution of precipitation coupled with extremely high temperatures was measured. Effects of global warming were experienced which manifested in weather extremities (heat stress, persistent drought, atmospheric aridity, hails, sudden floods). Overall, maize crops had to cope with extreme challenges, and heat-stressed maize showed the symptoms of rolling and water deficiency. Data for precipitation and air temperatures during the sampling period are shown in Figure 6.

### 5.4. Soil Nitrate Measurement

The efficiency of nitrapyrin was followed up with the measurement of nitrate content in the soil as it provides indirect information about the activity of nitrification. Soil samples were collected from the 0–30 cm depth, along a “W” line, taking into account soil heterogeneity. Samples were taken from nitrapyrin-free (CT, FF) blocks and nitrapyrin-treated (NP, NP + FF) blocks (six samples per treatment). In terms of nitrate changes in the soil, foliar fertilizer (FF) itself has no relevance because the amount of liquid that may leak into the soil is negligible. Foliar fertilization provides only a supplemental fertilization strategy that delivers nutrients directly to the above-ground plant organs [97]. Therefore, the foliar application must require sufficient leaf area to become effective [29]. Samplings were performed 7 times, approximately every 2–3 weeks, adapted to the prevailing weather conditions. Nitrate determination was conducted with a selective nitrate electrode (Nitrat 2000, Stelzner, Germany) following the recommended steps taken by the crop producer.

### 5.5. Relative Chlorophyll Measurement

Relative chlorophyll measurement was conducted at the V12 stage of maize (7 July 11th week after sowing) and at the R1 stage (silking, 2 August 14th week after sowing), using a SPAD-502 Plus Chlorophyll Meter (Konica Minolta, Japan). SPAD readings were observed on the top fully expanded leaf on five plants per treatment. On each leaf, five measurements were taken, and the mean values were retained.

### 5.6. Photosynthetic Pigment Quantification

A method reported by Moran and Porath [98] was adopted to determine individual photosynthetic pigments. Analysis of the results was used following Wellburn [99]. Fresh tissue samples were taken from the top fully expanded leaf of maize and stored in liquid nitrogen until measurement. 50 mg tissue samples were dissolved 5 mL *N*,*N*-dimethylformamide at 4 °C for 72 h. Extract absorbance was measured spectrophotometrically, which was adjusted at wavelengths of 470, 647, and 664 nm (Nicolet Evolution 300 UV-Vis Spectrometer; Thermo Fisher Scientific, Waltham, MA, USA). Five replications of crops per treatment were sampled to measure the photosynthetic pigment quantity.

### 5.7. MDA Measurement

A method described by Heath and Packer [100] was applied to measure the level of lipid peroxidation by determining the content of malondialdehyde (MDA). After grounding 0.1 g fresh leaf sample with liquid nitrogen, samples were homogenised in 1 mL solution containing 0.25% (*w*/*v*) thiobarbituric acid (TBA) and 10% (*w*/*v*) trichloroacetic acid (TCA). As a next step, samples were centrifuged at 10,800× *g* for 25 min at 4 °C. The obtained supernatant (0.2 mL) was transferred to clean Eppendorf tubes, in which 0.8 mL solution of 0.5% (*w*/*v*) TBA and 20% (*w*/*v*) TCA was prepared. The mixture was heated at 95 °C for 30 min in a termoshaker (Bioshan TS-100) and immediately cooled on ice. Absorbance measurements (Nicolet Evolution 300 UV-Vis Spectrometer) at 532 and 600 nm were used to measure the MDA concentration of samples. MDA determination was based on the extinction coefficient of 155 mM^−1^ cm^−1^. Five replications of crops per treatment were sampled to measure the rate of lipid peroxidation.

### 5.8. SOD Measurement

The activity of SOD was shown by the method described by Giannopolities and Ries [101], as well as Beyer and Fridovich [102], based on the ability of SOD to inhibit the photochemical reduction of nitroblue tetrazolium (NBT). Since NBT reduction is inversely proportional to enzyme activity, the SOD unit was defined as the amount of enzyme to induce 50% inhibition of the reduction of NBT. Following the standard, inhibition was monitored at 560 nm. Fresh top fully expanded leaf samples were cut and immediately were stored in liquid nitrogen until preparation. 0.4 g powdered leaf sample was homogenized in 4 mL 50 mM phosphate buffer (pH 7.8), which contained 1 mM phenylmethanesulfonyl fluoride (PMSF), 0.1 mM ethylenediaminetetraacetic acid (EDTA), and 1% (*w*/*v*) polyvinylpyrrolidone (PVP). Centrifugation of samples was adjusted 10,000× *g* for 15 min at 4 °C. Five replications of crops per treatment were sampled to measure the activity of SOD.

### 5.9. Proline Measurement

The proline content of leaf samples was determined based on the method described by Carillo and Gibon [103] with a few modifications. After grounding fresh leaf samples (0.3 g) with liquid nitrogen, 6 mL 70% (*v*/*v*) ethanol was added [104]. One mL reaction mixture of 1% ninhydrin in 60% (*v*/*v*) acetic acid was added into 500 μL ethanolic extract, which was transferred into 1.5 mL Eppendorf tubes and heated at 95 °C for 20 min. As a next step, samples were cooled and centrifuged at 12,000× *g* for 1 min. Spectrophotometric absorbance was read at 520 nm. Determination of proline content was calculated based on the proline standard curve. Five replications of crops per treatment were sampled to measure the proline content.

### 5.10. Laboratory Leaf Analysis

The efficacy of nitrapyrin and foliar fertilization treatments were monitored by laboratory leaf analysis, which provides accurate information about the current nutrient content of maize leaves. Twenty adult maize leaves (the nearest to the maize ear) were sampled in each treatment. Sampling was performed one week after the foliar fertilization (the 9th week after maize sowing), which is enough time for the applied nutrients to be utilized. Considering that the nutrient analysis demands sufficiently large tissues, the whole part of the sampled leaf was cut. The laboratory leaf analysis to determine the nutrient content of the treated maize crops was conducted by the HL-LAB Environmental and Soil Testing Laboratory (Debrecen, Hungary). As a first step, an SLW 240 drying oven (Pol-Eko-Aparatura, Wodzisław Śląski, Poland) was used for the preparation of samples from which extraction was performed with HNO_3_-H_2_O_2_ blend with MARS 6 microwave digester (CEM Corporation, Matthews, NC, USA). The N content of the samples was determined on a UDK 139 Semi-Automatic Kjeldahl Distillation Unit (VELP Scientifica, Usmate, Italy), while the other elements were determined using an iCAP 6300 Radial View ICP-OES spectrometer (Thermo Fisher Scientific, USA).

### 5.11. Biomass Productivity

The effectiveness of nitrapyrin as a nitrogen-enhancing treatment was monitored by measuring the root mass and stem diameter of maize, as these two parameters are a good indicator of biomass production. Biomass production is primarily determined by the N-availability in the soil and N-use efficiency of maize [88]. Root samplings and stem diameter measurements were taken only from nitrapyrin-containing treatments (NP, NP + FF) and from the control plot (CT) and were not obtained separately from foliar fertilizer treatment (FF), as its main purpose is to supply other micronutrients or to make up for nutrient deficiencies. For root sampling, to avoid bias due to the different sample sizes, 30 cm^3^ of soil was dug out from the surface and root samples (10 roots per treatment) were cleaned from soil residues and dried at 60 °C until constant weight. As a next step, the weight of clean and dried root samples was measured. Stem diameter (20 per treatment) was measured using a caliper above the three lowest nodes. The mean of the three measurements was retained as a single value. Samplings and measurements were conducted on September 9, nearing the end of the growing season.

### 5.12. Parameters of Maize Yield

Of the different maize yield parameters, the length and diameter of the ear were measured. In addition, the number of rows in each ear, kernel number per row, and 1000-kernel weight were also determined. The quality parameters of yield, including the protein, oil, and starch content of kernels were determined using Infratech 1241 Grain Analyzer (FOSS, Hilleroed, Denmark).

### 5.13. Statistical Analysis

The statistical analysis was conducted using the R programming language (v4.1.1) [105] and the “agricolae” R package (v1.3-5) [106]. Student’s *t*-test and its non-parametric version (Mann-Whitney U test) and one-way ANOVA with Duncan’s multiple range test were performed to examine the means at a significance level of 5%.

## Figures and Tables

**Figure 1 plants-10-02426-f001:**
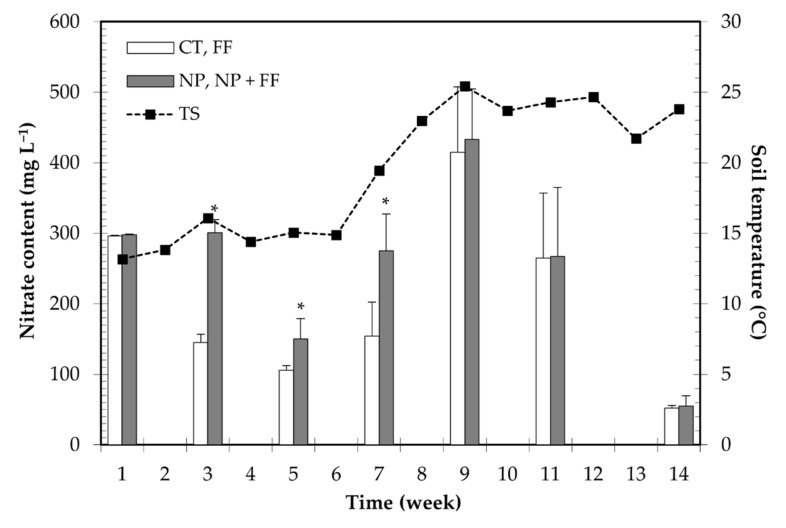
Nitrate content changes in the soil during the growing season due to nitrapyrin treatment and soil temperature. Each column represents the means ± SD (n = 6) of nitrate measurements. Data of the dashed line show the weekly mean of average daily soil temperature. CT: untreated control; FF: foliar fertilizer; NP: nitrapyrin-treated soil; NP + FF: joint treatment of nitrapyrin and foliar fertilizer; TS: soil temperature; Asterisks (*) indicate significant differences (Student’s *t*-test, *p* < 0.05) between nitrapyrin-treated and untreated soils.

**Figure 2 plants-10-02426-f002:**
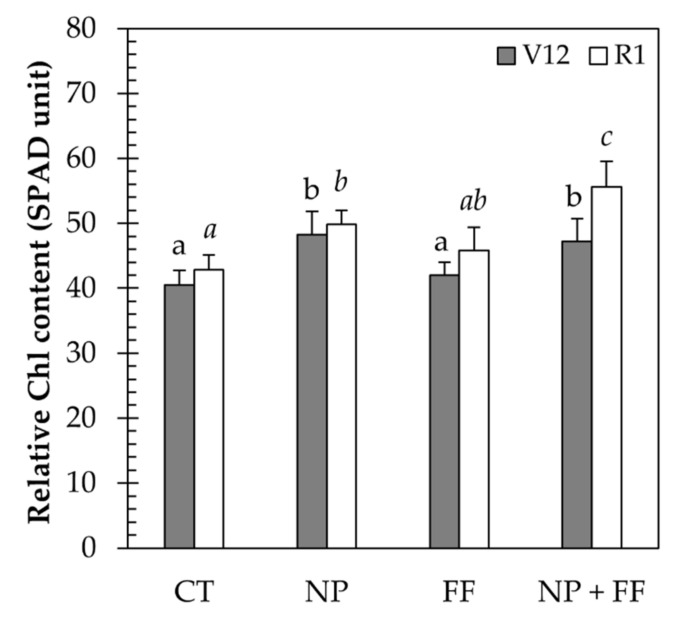
The relative chlorophyll (Chl) content of maize leaves due to each treatment at different phenological stages (V12 and R1). Data represent mean ± SD (n = 25). CT: untreated control; FF: foliar fertilizer; NP: nitrapyrin; NP + FF: joint treatment of nitrapyrin and foliar fertilizer. Different lowercase letters denote significant differences (one-way ANOVA and Duncan’s multiple range test, *p* ≤ 0.05) between the different treatments.

**Figure 3 plants-10-02426-f003:**
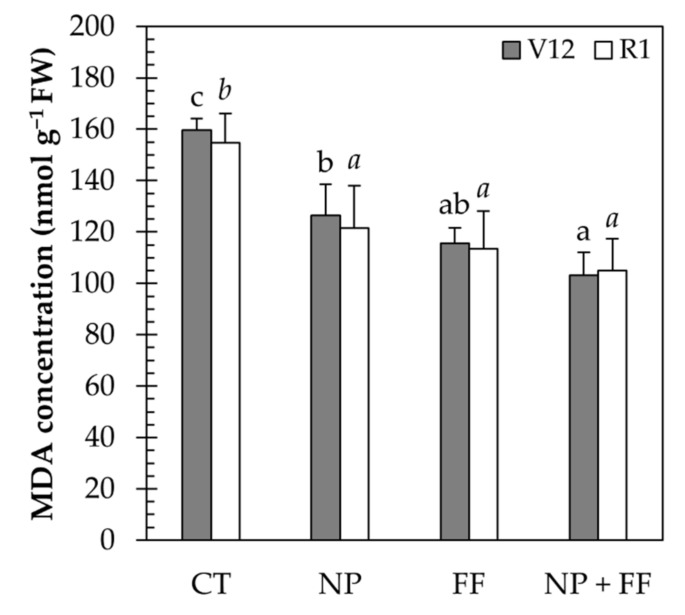
Changes in MDA concentration of maize leaves due to each treatment at different phenological stages (V12 and R1). Data represent mean ± SD (n = 5); CT: untreated control; FF: foliar fertilizer; NP: nitrapyrin; NP + FF: joint treatment of nitrapyrin and foliar fertilizer. Different lowercase letters denote significant differences (one-way ANOVA and Duncan’s multiple range test, *p* ≤ 0.05) between the different treatments.

**Figure 4 plants-10-02426-f004:**
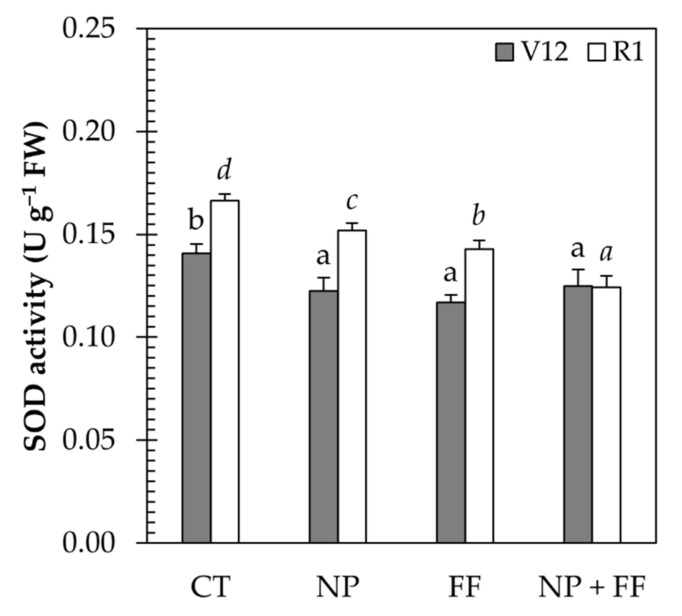
SOD activity osf maize leaves due to each treatment at different phenological stages (V12 and R1). Data represent mean ± SD (n = 5); CT: untreated control; FF: foliar fertilizer; NP: nitrapyrin; NP + FF: joint treatment of nitrapyrin and foliar fertilizer, FW: fresh weight. Different lowercase letters denote significant differences (one-way ANOVA and Duncan’s multiple range test, *p* ≤ 0.05) between the different treatments.

**Figure 5 plants-10-02426-f005:**
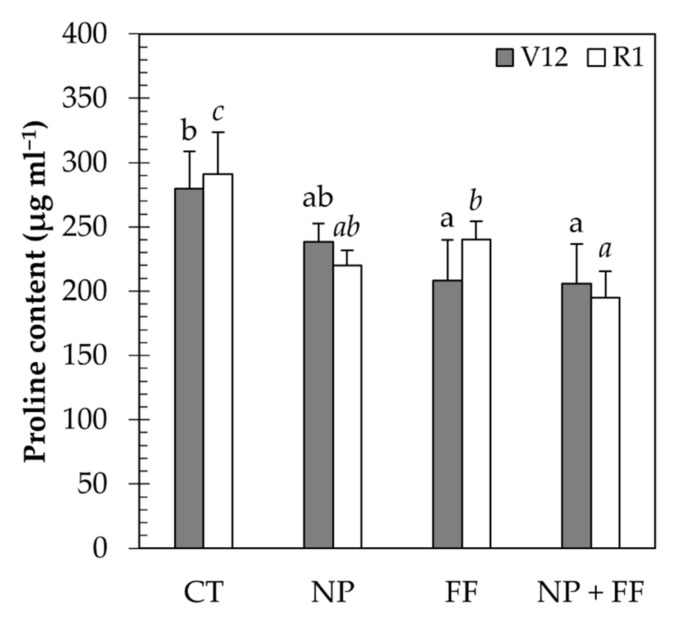
Proline content of maize leaves due to each treatment at different phenological stages (V12 and R1). Data represent mean ± SD (n = 5); CT: untreated control; FF: foliar fertilizer; NP: nitrapyrin; NP + FF: joint treatment of nitrapyrin and foliar fertilizer. Different lowercase letters denote significant differences (one-way ANOVA and Duncan’s multiple range test, *p* ≤ 0.05) between the different treatments.

**Figure 6 plants-10-02426-f006:**
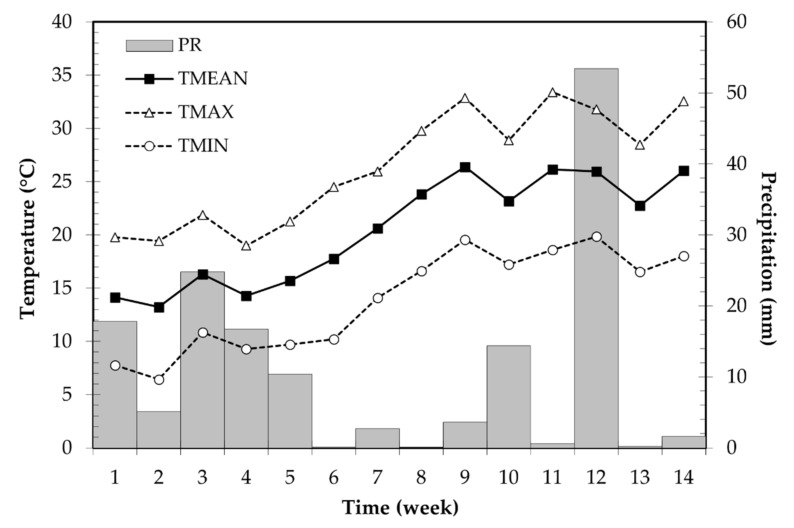
Meteorological data of growing season in the study area. PR: precipitation; TMEAN: mean daily temperature; TMAX: mean daily maximum temperature; TMIN: mean daily minimum temperature.

**Figure 7 plants-10-02426-f007:**
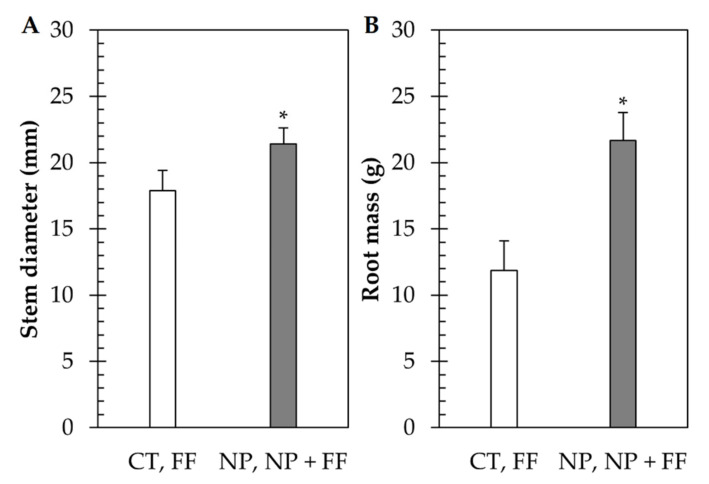
Biomass productivity of maize in different treatments. (**A**) Stem diameter; (**B**) Root mass. Data represent mean ± SD (n = 5); CT: untreated control; FF: foliar fertilizer; NP: nitrapyrin; NP + FF: joint treatment of nitrapyrin and foliar fertilizer. Asterisks (*) indicate significant differences (Student’s *t*-test, *p* < 0.05) between nitrapyrin-treated and control plants.

**Table 1 plants-10-02426-t001:** The photosynthetic pigments of maize leaves as a result of each treatment at different phenological stages (V12 and R1).

Stage	Treatments	Chl *a* (mg g^−1^ FW)	Chl *b* (mg g^−1^ FW)	Chl *a*/*b* Ratio	Car (mg g^−1^ FW)	Total Chl/Car Ratio
V12	CT	17.76 ± 0.87 ^a^	4.89 ± 0.52 ^a^	3.67 ± 0.50 ^a^	13.44 ± 0.82 ^a^	1.69 ± 0.08 ^a^
	NP	19.12 ± 0.27 ^b^	5.28 ± 0.26 ^a^	3.63 ± 0.22 ^a^	12.52 ± 0.96 ^a^	1.96 ± 0.14 ^b^
	FF	18.61 ± 0.56 ^a, b^	4.88 ± 0.59 ^a^	3.87 ± 0.61 ^a^	13.22 ± 0.80 ^a^	1.78 ± 0.08 ^a, b^
	NP + FF	19.84 ± 0.87 ^b^	5.70 ± 0.88 ^a^	3.55 ± 0.70 ^a^	13.58 ± 0.36 ^a^	1.88 ± 0.06 ^b^
R1	CT	18.48 ± 1.04 ^a^	4.81 ± 0.16 ^a^	3.84 ± 0.19 ^a^	13.20 ± 0.65 ^a^	1.76 ± 0.08 ^a^
	NP	19.91 ± 0.65 ^b^	5.39 ± 0.73 ^a^	3.75 ± 0.63 ^a^	12.80 ± 0.39 ^a^	1.98 ± 0.09 ^b^
	FF	19.72 ± 0.30 ^b^	5.19 ± 0.56 ^a^	3.83 ± 0.45 ^a^	13.89 ± 0.46 ^a, b^	1.80 ± 0.08 ^a, b^
	NP + FF	21.17 ± 0.15 ^c^	6.68 ± 0.68 ^b^	3.19 ± 0.32 ^a^	14.80 ± 0.78 ^b^	1.89 ± 0.13 ^a, b^

Data represent mean ± SD (n = 5); Chl: chlorophyll; Car: carotenoids; CT: untreated control; NP: nitrapyrin; FF: foliar fertilizer; NP + FF: joint treatment of nitrapyrin and foliar fertilizer, FW: fresh weight. Different lowercase letters denote significant differences (one-way ANOVA and Duncan’s multiple range test, *p* ≤ 0.05) between the different treatments.

**Table 2 plants-10-02426-t002:** The nutrient content of maize leaves due to each treatment. All values in mg kg^−1^ of air-dried substance; Different lowercase letters indicate values that are less than the lower critical value of the optimal content: ^a^—28,000 mg kg^−1^ [46], ^b^—2500 mg kg^−1^ [46], ^c^—17,000 mg kg^−1^ [46], ^d^—2100 mg kg^−1^ [46]; CT: untreated control; NP: nitrapyrin; FF: foliar fertilizer; NP + FF: nitrapyrin and foliar fertilizer.

Treatments	CT	NP	FF	NP + FF
Nitrogen (N)	24,100 ± 1205 ^a^	24,100 ± 1205 ^a^	22,900 ± 1145 ^a^	24,900 ± 1245 ^a^
Phosphorus (P)	2120 ± 85 ^b^	2240 ± 90 ^b^	2120 ± 85 ^b^	2450 ± 98
Potassium (K)	10,080 ± 403 ^c^	10,160 ± 406 ^c^	11,640 ± 466 ^c^	13,150 ± 526 ^c^
Calcium (Ca)	6200 ± 465	7350 ± 551	5900 ± 443	5580 ± 419
Magnesium (Mg)	8400 ± 630	7940 ± 596	7190 ± 539	6570 ± 493
Sulphur (S)	1700 ± 170 ^d^	1870 ± 187 ^d^	1750 ± 175 ^d^	1710 ± 171 ^d^
Boron (B)	40.5 ± 5.1	33.3 ± 4.2	43.4 ± 5.4	30.0 ± 3.8
Copper (Cu)	8.1 ± 0.6	12.1 ± 0.9	12.7 ± 1.0	9.1 ± 0.7
Iron (Fe)	189.5 ± 14.2	123.8 ± 9.3	168.1 ± 8.4	108.0 ± 5.4
Manganese (Mn)	63.5 ± 4.8	99.3 ± 7.4	75.7 ± 5.7	75.5 ± 5.7
Zinc (Zn)	30.1 ± 2.3	26.1 ± 2.0	33.6 ± 2.5	23.5 ± 1.8

**Table 3 plants-10-02426-t003:** Parameters of maize yield. Data represent mean ± SD (n = 10); Chl: chlorophyll; Car: carotenoids; CT: untreated control; NP: nitrapyrin; FF: foliar fertilizer; NP + FF: joint treatment of nitrapyrin and foliar fertilizer, FW: fresh weight. Different lowercase letters denote significant differences (one-way ANOVA and Duncan’s multiple range test, *p* ≤ 0.05) between the different treatments.

Treatments	CT	NP	FF	NP + FF
1000 Kernel weight (g)	346	475	398	404
Cob length (cm)	15.22 ± 1.15 ^a^	22.12 ± 1.38 ^c^	18.83 ± 2.17 ^b^	22.18 ± 1.19 ^c^
Cob diameter (mm)	42.92 ± 5.11 ^a^	49.71 ± 2.40 ^b^	45.85 ± 3.65 ^a^	50.89 ± 1.80 ^b^
Number of rows per cob	14.8 ± 1.0 ^a^	16.2 ± 1.5 ^b^	15.8 ± 1.1 ^a, b^	16.0 ± 0.9 ^b^
Number of grains per row	33.3 ± 5.9 ^a^	42.2 ± 2.7 ^c^	32.2 ± 3.2 ^a^	38.2 ± 4.9 ^b^
Moisture (%)	21.4	21.6	21.2	21.7
Starch content (%)	53.7	53.8	54.7	54.2
Oil content (%)	4.6	4.6	4.3	4.5
Protein content (%)	8.7	9.1	8.9	9.3

**Table 4 plants-10-02426-t004:** Composition of the applied foliar fertilizer.

Nutrient	mg L^−1^
Total nitrogen (N) ^a^	80,000
Phosphoric anhydride (P_2_O_5_)	70,000
Potassium oxide (K_2_O)	70,000
Sulphur trioxide (SO_3_)	70,000
Boron (B) ^b^	127
Copper (Cu) ^b^	162
Iron (Fe) ^b^	284
Manganese (Mn) ^b^	265
Zinc (Zn) ^b^	38

^a^—urea nitrogen; ^b^—EDTA chelated.

## Data Availability

The data presented in this study are available upon request from the corresponding author.

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
