# Peer review of "Examination of the Productivity and Physiological Responses of Maize (Zea mays L.) to Nitrapyrin and Foliar Fertilizer Treatments"

_plants, 2021, doi:10.3390/plants10112426_

Round 1

Reviewer 1 Report

Dear authors:

Extensive changes have been made to the article, however, some minor corrections must be made to accept the paper for publishing. Examples are:

  1. line 52:
  2. Sentence: line 61-62
  3. There are abreviations in Figures that are not explained in the legends. FF: foliar fertilizer in figure 1.
  4. In figure 1 the levels of Nitrate in the soil treated with Nitrapyrin are always higher than the control. The authors explain in the text "Results suggest that the inhibitory effect of nitrapyrin on nitrification was significant until the seventh week after the nitrapyrin treatment, as the nitrate content in the nitrapyrin-treated (NP) soil was much lower than in the control field plot"  They are never lower than the control.

This must be discussed in the Discusion section.

Reviewer 2 Report

I would like to thank all the authors for their efforts and thoroughness in improving their manuscript. I have no more comments for authors.

Reviewer 3 Report

In the manuscript entitled "Examination of the Productivity and Physiological Responses of Maize (Zea mays L.) to Nitrapyrin and Foliar Fertilizer Treatments" authors describe applying Nitrapyrin and foliar fertilizers to increase the amount of available nitrogen. Nitrapyrin is a well-known nitrification inhibitor, which has a positive effect on the efficiency of the use of nutrients by plants and on their development. In this work, the main indicators of plant stress were measured in the tissues of maize leaves: proline, malondialdehyde (MDA), relative chlorophyll, photosynthetic pigments and superoxide dismutase (SOD) activity. In addition, the accumulation of biomass, as well as the quantitative and qualitative parameters of the yield were studied.

In this manuscript, the authors logically plan their experiments. However, this work is not without its drawbacks, which I describe below. Before this manuscript can be accepted for publication, it is necessary to improve this work.

  • In table 2, you present the data on the content of some elements in maize plants that are under stress conditions (drought). In this regard, the data are not entirely correct and questions arise, since the rest of the analyzes were carried out on plants under optimal conditions (if I understood correctly). Why there is an increase in Cu when NP and FF are treated, and when combined, the effect disappears? Also, when processing NP and FF, a decrease in Fe content is observed.
  • Were untreated plants used as a control? Spraying with plain water will also affect various physiological and biochemical parameters of the plants. That is, the use of water in the control would be more correct, since Nitrapyrin was also dissolved in water.
  • The authors constantly use the word " significantly" in the results to emphasize the statistical significance, but this has a negative effect on the style of the text. I suggest using " significantly" only to highlight the most striking results.
  • You suggest that nitrapyrin contributed to retaining more available N-forms in the soil, resulting in maize crops increasing their biomass production. However, this statement contradicts the data from Table 1, where a large increase is not observed.
  • The main thing I miss is the novelty of the work. Since there are works in which such experiments were done, I would like the authors to describe this moment more specifically.

Line 55-56. “as a result” this wording is used 2 times in one sentence. Please rephrase the sentence.

Line 108-112. I think this proposal can be removed.

Line 130-132. I don't like the writing of this sentence. It is not entirely clear what you want to say. Repeats "relative" twice.

Line 301. You are talking about an increase in chlorophyll B, but judging by table 1, it was not reliable.

Line 305. How do you explain the synergy effect of NP and FF?

Line 132. "Results obtained" Remove italics.

Round 2

Reviewer 3 Report

Thanks for the detailed answer. I am satisfied with this version of the manuscript.

This manuscript is a resubmission of an earlier submission. The following is a list of the peer review reports and author responses from that submission.

Round 1

Reviewer 1 Report

The authors must change figure 1. It is a duplicate of the material and methods figure.

What is the central question addressed by the research? 
This study wanted to found a correlation between nitrogen fertilization (soil and foliar) in the field and specific stress indicators at two different growth stages. However, the authors did not identify which stress the crop was suffering at the time point of sample collection. The stress indicators used are indeed quite common and give some information but not specific information. They missed or avoided growth, nutritional analysis and productivity information in this paper.
Is it relevant and interesting? No, it is not because they did not think to include this paper in a hot topic (Biostimulants, abiotic stress response, etc..) engaging the reviewers.
How original is the topic?  It is not Original.
What does it add to the subject area compared with other published material? 
Is the paper well written? Yes, it is.
Is the text clear and easy to read? Yes, It is easy to read.
Are the conclusions consistent with the evidence and arguments presented? Yes, but the paper needs more information from the pieces of evidence to address the conclusion.
Do they address the main question posed? They address the question partially. There are mistakes in the results with figure 1, and as was mentioned before, they missed or avoided growth, nutritional and productivity data from a crop growing under field conditions.

Generally, It is not a bad paper but with whole new writing and focusing on Nitrapyrin [2-chloro-6-(trichloromethyl)-pyridine] as a promising strategy to enhance the efficiency of N utilization. An d the results presented here must be compleated answering questions such as: 
Could be these Nitrogen stabilizing products be considered Biostimulants? 
How do they affect soil bacteria or plant growth-promoting bacteria?
There is an improvement in plants growth, nutritional status and productivity?
In brief, I will not reject this paper If the authors are committed to extensive work on it.

Reviewer 2 Report

Authors in presented work aimed to show the importance of N supply to improve the stress tolerance of maize. Idea is based on the fact that nitrogen supply can contribute to tolerance of plants and the response of plants was evaluated certain stress response parameters.

Sampling was done in weather conditions that correspond to the drought period.  In the discussion (line 332-333) authors state that physiological symptoms of drought were present but they didn’t show any of the physiological parameters (biomass, hight, number of leaves and size, relative water content,  pitcures of plants) to support the hypothesis that treatments (nitrapyrin or foliar fertilizers) improved tolerance of maize.

Treatment triggered metabolic changes in common stress responsive processes (treatments always trigger response in certain manner) that can be described as improved tolerance but without strong growth related parameters study is superficial and descriptive.

In addition, figure 1 caption and figure 1 (line 112) doesn’t show nitrate content changes in soil what is stated but precipitation and temperature fluctuations. The same figure is repeated as Figure 6 (line 335) and placed under discussion 4.4.

Reviewer 3 Report

All my comments are in the attached file.
